# Inequalities in Out-of-Pocket Health Expenditure Measured Using Financing Incidence Analysis (FIA): A Systematic Review

**DOI:** 10.3390/healthcare12101051

**Published:** 2024-05-20

**Authors:** Askhat Shaltynov, Ulzhan Jamedinova, Yulia Semenova, Madina Abenova, Ayan Myssayev

**Affiliations:** 1Epidemiology and Biostatistics Department, Semey Medical University, Semey 071400, Kazakhstan; ulzhan.jamedinova@smu.edu.kz (U.J.); madina.abenova@smu.edu.kz (M.A.); 2School of Medicine, Nazarbayev University, Astana 010000, Kazakhstan; yuliya.semenova@nu.edu.kz; 3Department of the Science and Human Resources, Ministry of Healthcare of the Republic of Kazakhstan, Astana 010000, Kazakhstan; a.myssayev@dsm.gov.kz

**Keywords:** out-of-pocket health expenditures, financing incidence analysis, universal health coverage, catastrophic health expenditures, health inequality

## Abstract

Government efforts and reforms in health financing systems in various countries are aimed at achieving universal health coverage. Household spending on healthcare plays a very important role in achieving this goal. The aim of this systematic review was to assess out-of-pocket health expenditure inequalities measured by the FIA across different territories, in the context of achieving UHC by 2030. A comprehensive systematic search was conducted in the PubMed, Scopus, and Web of Science databases to identify original quantitative and mixed-method studies published in the English language between 2016 and 2022. A total of 336 articles were initially identified, and after the screening process, 15 articles were included in the systematic review, following the removal of duplicates and articles not meeting the inclusion criteria. Despite the overall regressivity, insurance systems have generally improved population coverage and reduced inequality in out-of-pocket health expenditures among the employed population, but regional studies highlight the importance of examining the situation at a micro level. The results of the study provide further evidence supporting the notion that healthcare financing systems relying less on public funding and direct tax financing and more on private payments are associated with a higher prevalence of catastrophic health expenditures and demonstrate a more regressive pattern in terms of healthcare financing, highlighting the need for policy interventions to address these inequities. Governments face significant challenges in achieving universal health coverage due to inequalities experienced by financially vulnerable populations, including high out-of-pocket payments for pharmaceutical goods, informal charges, and regional disparities in healthcare financing administration.

## 1. Introduction 

Access to healthcare is a fundamental human right, and financial protection is critical in ensuring equitable and affordable access to healthcare. Universal health coverage (UHC) is a key mechanism for providing financial protection in healthcare, and it has been recognized as a key target in the Sustainable Development Goal (SDG) 3.8 adopted by the United Nations in 2015. Despite efforts to achieve UHC, out-of-pocket (OOP) health expenditures remain a significant financial burden for many individuals and households globally [1].

In their article published in the Bulletin of the World Health Organization (WHO), Barber et al. identified four challenges for financing the WHO’s goals and primary healthcare: 1. Global normative expenditure targets were primarily devised for advocacy purposes, emphasizing the significance of healthcare for national development and securing political commitment. 2. Focusing attention on global normative targets may lead to the mistaken assumption that achieving universal health coverage (UHC) is a fixed threshold or singular, unchanging goal over time. 3. The concept of a global normative target assumes that all countries need to spend a certain amount on healthcare to achieve similar outcomes. 4. Global normative targets typically focus attention solely on funding deficits, leading some policymakers and donors to assume that private financing could fill the gap left by limited government budgetary capacity. Limited budgets in low-income countries can lead to unrealistic assessments when comparing health spending with global goals. UHC is a goal that requires health system reform and investment, and the efficiency of spending matters more than the amount spent on health to achieve goals. The use of private financing and private insurance schemes often contradicts the WHO’s goals and financial protection [2]. While private funding improves healthcare financing, high OOP spending does not contribute to UHC and increases inequality [3]. 

Several studies demonstrate the significance of UHC in ensuring financial protection in healthcare. For instance, a study in Thailand showed that policy aimed to achieve UHC reduced catastrophic healthcare expenditures (CHEs) from 6 to 2%, considerably alleviating financial burdens on households [4]. In Burkina Faso, interventions related to UHC reduced inequalities in healthcare spending distribution and improved access to healthcare services, especially for the poor [5]. These findings highlight the importance of UHC-achieving policy in providing financial protection and improving access to healthcare services. 

UHC is achieved through a combination of mechanisms, including health insurance, social health protection, and community-based health financing. UHC has been recognized as a key target in the SDGs, with SDG 3.8 specifically calling for “achieving universal health coverage, including financial risk protection”. Joseph Kutzin highlighted in his publication that equity or fairness in healthcare financing, where households contribute to the healthcare system based on their ability to pay, should be a vital objective of healthcare systems to support UHC [6].

In recent years, there has been an increasing interest in understanding the impact of health financing on individuals’ financial protection and equity in healthcare access [7,8]. The use of FIA has been shown to be particularly valuable in identifying the extent of inequalities in OOP health expenditure [9]. Previous studies have highlighted the existence of significant inequalities in OOP health expenditure across different income groups, with the poorest populations often bearing a disproportionate burden of health financing [10,11]. These findings have important implications for policymakers, as they highlight the need for policies aimed at improving financial protection for vulnerable populations.

A range of international studies evaluated the impact of vertical inequality in OOP payments on healthcare, and there are two systematic reviews on the topic [12,13]. However, previous reviews on this matter are based on studies published before 2016, while the situation with universal health service coverage, particularly household OOP expenditures, requires continuous monitoring until 2030. The aim of this systematic review was to assess OOP health expenditure inequality measured by the FIA across different territories, in the context of achieving UHC by 2030.

## 2. Methods

### 2.1. Search Strategy 

The following databases were searched: PubMed, Scopus, and Web of Science. 

The search strategy for this study involved a set of predefined key terms, including the following terms: (inequality OR equity OR inequit* OR equit* OR progressiv* OR regressiv*) AND (health OR healthcare OR health care) AND (expenditure OR payment OR spending OR financ* OR cost) AND (OOP OR out-of-pocket OR pocket OR private) AND (Kakwani OR FIA OR financing incidence analysis).

We did not set a specific target value for the number of publications included in the review. The search strategy and inclusion and exclusion criteria were designed to capture all existing evidence in the databases using Boolean operators, synonyms, and alternative terms in the search query.

A publication period filter was defined from 2016 to 2022.

### 2.2. Study Inclusion Criteria

Original quantitative and mixed-method studies published in the English language between 2016 and 2022 were included in the review if they consisted of financing incidence analysis (FIA) and calculated Kakwani index of inequalities in health OOP expenditures. Qualitative studies, systematic reviews, and studies without full text were excluded.

### 2.3. Data Extraction

In the first stage of the study, Askhat Shaltynov transferred the identified studies to the Rayyan.ai website and removed any duplicates. Subsequently, Askhat Shaltynov and Ulzhan Jamedinova independently evaluated the titles and abstracts of the remaining articles to determine their suitability for inclusion. Any articles deemed irrelevant were eliminated from consideration.

The authors proceeded to collect data from the selected studies and recorded them in a data extraction sheet. Askhat Shaltynov, Ulzhan Jamedinova, and Madina Abenova were responsible for gathering the information, while the other authors reviewed and verified the data. The extracted information included details such as the authors’ names, publication year, country of origin, sample size, urban or rural location, whether the data were collected from households or individuals, and equity indicators reported in the studies.

Any disagreements between the reviewers were resolved through discussions with two additional team members. 

To provide transparency in the selection process, the authors created the Preferred Reporting Items for Systematic Reviews and Meta-Analyses (PRISMA) checklist (Appendix A) in gathering the relevant information.

### 2.4. Risk of Bias (Quality) Assessment

The evaluation of the included articles was conducted using the Strengthening the Reporting of Observational Studies in Epidemiology (STROBE) tool. To assess the quality of the full-text articles, we evaluated the sources of information for each study, the quality of the statistical analysis, and the description of the sample (number of participants, households, etc.). The evaluation was conducted by Askhat Shaltynov and Ulzhan Jamedinova.

### 2.5. Data Analysis and Synthesis

We evaluated measures of inequality by grouping study results according to the Kakwani index, which ranges from regressive (−2) to progressive (1). In addition, for studies that provided data on CHE, we assessed the incidence of this measure. We also assessed studies reporting on healthcare financing reforms implemented by countries.

Our approach allowed for a comprehensive evaluation of measures of inequality across a range of studies. By grouping study results based on the Kakwani index, we were able to analyze the distribution of healthcare expenditures and identify trends in financing. Furthermore, by assessing the incidence of CHE, we were able to gain insights into the financial burden imposed on households due to healthcare costs.

The systematic review protocol was published in the PROSPERO database, ID: CRD42023395936.

### 2.6. Map Visualization 

The map of study locations developed in the ArcGIS PRO 3.0 software used the ESRI base map and vector layers of continents and country borders. 

### 2.7. Definitions Used

Universal health coverage (UHC) refers to a system where all individuals and communities can access quality healthcare services without suffering financial hardship.

Financing incidence analysis (FIA) is a tool that allows for the measurement of the distributional impact of health financing mechanisms on different income groups [14].

Catastrophic health expenditures (CHEs) are those healthcare expenses that surpass 40% of the total consumption of healthcare services. CHEs can be defined as OOP expenditures that exceed 5%, 10%, 20%, and 25% of total consumption or income [15]. The WHO and World Bank have defined CHE for SDG 3.8.2 using a budget threshold of 10% or 25% of the total budget [16].

The Kakwani index is a measure commonly used to assess the progressivity of healthcare financing. The Kakwani index is particularly useful in identifying the extent to which different income groups benefit from government health subsidies and assessing the progressivity of OOP health expenditure [17,18]. The Kakwani index is defined as twice the area between the concentration curve and the line of equality, divided by the mean OOP health expenditure. It can take values between −2 and 1. A positive value indicates that the health financing mechanism is progressive, indicating that wealthier households contribute a higher proportion relative to their share of total household income. Conversely, a negative value suggests that the health financing mechanism is regressive, meaning that poorer households contribute a larger proportion of their income compared to their share of total household income.
K = C − G, (1)
where K is the Kakwani index; C is the concentration index for healthcare spending; G is the Gini coefficient for household income (ability to pay).

### 2.8. Registration

The systematic review protocol was registered in the PROSPERO database, ID: CRD42023395936.

## 3. Results

### 3.1. The Selection of Studies

A total of 336 articles were retrieved from various databases. Following the removal of 77 duplicates, 259 articles were screened based on their titles and abstracts. Subsequently, 85 articles were subjected to full-text analysis. Of these 85 articles, 69 were excluded for not meeting the inclusion criteria. Finally, 16 articles were selected for a detailed examination as part of a systematic review (Figure 1) [19].

A list of these publications, their respective data sources, and their reported metrics is provided in Table 1. These publications encompassed data collected from a diverse range of countries (Figure 2).

Notably, the highest frequency of analyses was observed for data collected from Bangladesh (n = 2), China (n = 3), and Iran (n = 3). 

The inclusion of a diverse set of countries in the review adds to the comprehensiveness of the topic. 

Based on the analysis of the included studies, it is evident that the research was conducted at various levels, including national and subnational/regional.

A substantial number of studies, specifically 10 in total, were conducted at the national level, with 9 of them being based on secondary data from national household expenditure surveys. These studies aimed to capture a broad view of the entire population or a representative sample of the country. In terms of subnational or regional research, there were six studies conducted specifically focusing on various regions within a country. It is noteworthy that the subnational survey conducted in Canada relies on secondary data obtained from the national Canada’s Survey of Household Spending. The other five surveys, encompassing both national and regional levels, including three from China, are grounded in primary data collection.

Seven out of the total number of studies included in the review exclusively examined inequality in OOP expenditures only, while others explored OOP expenditures and additional mechanisms of healthcare financing. Furthermore, among the ten studies, information regarding CHE was provided in six of them. All sixteen studies utilized the Kakwani index as an indicator of FIA to analyze inequality in OOP expenditures.

Figure 3 presents data on OOP healthcare expenditure from various studies conducted in high-income countries and low- and middle-income countries. The Kakwani index, used to measure the progressivity or regressivity of healthcare financing, reveals important findings. The study conducted by Sarker et al. in Bangladesh reported the lowest Kakwani index value, indicating a regressive nature of healthcare financing in that context [20]. Conversely, the study by Abdi et al. conducted in Iran showed the highest Kakwani index value, indicating a progressive healthcare financing system with a value of 0.15 [21]. Additional regions covered in the figure include Canada, China, India, Italy, Mauritius, South Korea, Portugal, and Turkey. 

Table 2 presents data on the Kakwani index as reported in the selected articles, covering the most recent year across all funding sources. China, Mauritius, and South Korea show a slight progressivity (Kakwani index is close to 0) in direct taxes. This means that individuals with higher incomes contribute slightly more in taxes than those with lower incomes. Direct taxes are those imposed directly on individuals or entities, such as income taxes or property taxes, while indirect taxes are imposed on goods and services, like sales tax or value-added tax. Direct payments refer to payments made directly by individuals or entities for healthcare services, while indirect payments refer to payments made indirectly through taxes or insurance premiums. Other countries do not provide specific data on the progressivity of direct taxes. With the exception of South Korea, the selected studies consistently showed a negative value of the Kakwani index in the distribution of social insurance. This suggests that individuals with lower incomes in these countries spend a higher percentage of their money on social insurance compared to those with higher incomes.

### 3.2. South and Southeast Asia Region

Three studies were selected from the Southeast Asia region, specifically conducted in Bangladesh and Malaysia, to investigate the issue of inequality in OOP health expenditures. One study from South Asia was conducted in India. These articles offer valuable insights into the topic at hand. 

Sarker et al. (2021) conducted a study using data from the Household Income and Expenditure Survey (HIES) 2016 in Bangladesh [20]. They employed the Kakwani index to analyze the distribution of OOP payments among different socioeconomic groups, and their findings revealed that the poorest quintile in Bangladesh experienced a disproportionately higher burden of OOP payments for healthcare. Additionally, it was observed that on average, households spent 7.7% of their income on healthcare expenses, while the poorest households allocated 35% of their total income, indicating a highly regressive Kakwani index for OOP healthcare expenditures. It is worth noting that 32% of these OOP expenses were financed through borrowing by the population. 

Similarly, Molla et al. (2017) utilized data from the Bangladesh HIES 2010 to assess progressivity patterns of OOP healthcare expenditures and indicated that the overall volume of OOP healthcare expenditures can be disaggregated into different quintiles, with the poorest quintile accounting for 13.4%, the second quintile for 17.8%, the middle quintile for 22%, the fourth quintile for 21.3%, and the richest quintile for 25.5% [22]. Despite the fact that wealthier individuals contribute a larger absolute amount to healthcare financing compared to the poor, OOP payments are predominantly concentrated among the lower socioeconomic strata of the population. On the whole, the proportion of OOP healthcare expenditures in the payments of households in Bangladesh accounted for 63%, while social insurance only covered the formal sector.

Additionally, Mohamed Fakhri Abu Baharin et al. (2022) examined the equity of OOP payments for healthcare in Malaysia, utilizing data from the Household Expenditure Survey (HES) 2014/2015 [23]. This study focuses exclusively on OOP healthcare expenditures and provides an in-depth examination of expenditure categories within this domain. The distribution of household expenditures and OOP healthcare payments varied across different quintiles of household average total expenditure. The wealthiest quintile (Q5) accounted for nearly half the proportion of total household expenditures (42.06%), whereas the poorest quintile (Q1) represented less than 10% of this share (7.72%). A similar trend was observed for OOP healthcare payments, with Q5 having the largest share (46.98%) compared to Q1, which had a significantly smaller share, 4.91%. Both household expenditures and OOP healthcare payments exhibited a higher concentration among wealthier households, as evident from the positive concentration index value of 0.4296 for OOP healthcare payments. According to the authors, the progressive nature of this phenomenon can be explained by the fact that wealthier households, who can afford it, generally prefer to seek treatment at more expensive private healthcare facilities, consequently leading to lower-income households, such as those with moderate and low levels of income, relying more on government-subsidized and comparatively cheaper public healthcare services in Malaysia.

The study encompassing various districts in India revealed a consistent pattern of declining OOP healthcare expenditures as one transitions from lower-income to higher-income population quintiles [24]. This trend was observed across all eight districts examined, suggesting a regressive nature and potential burden of healthcare expenses borne by disadvantaged households in comparison to their more affluent counterparts. These findings were corroborated by negative Kakwani index values observed in all districts. The findings showed that the rural sample experienced a higher catastrophic burden compared to the urban one across all districts, with the districts of Kanpur Dehat and Meerut in Uttar Pradesh having the highest proportion of individuals facing this burden. These disparities could be attributed, in part, to variations in disease profiles and healthcare-seeking behaviors, as well as discrepancies in the healthcare supply landscape, including the mix of providers from the public and private sectors.

Overall, these articles provide valuable evidence of the inequality in OOP health expenditures in the South and Southeast Asia region, with variations observed between India, Bangladesh, and Malaysia.

### 3.3. East Asia

Several studies were identified that examined the equity of healthcare financing in China and South Korea, with a particular focus on the distributional impact of OOP payments. 

In China, the study by Chen et al. (2017) analyzed the distribution of household expenditures and healthcare payments in North Jiangsu in 2012, focusing on the progressivity of different healthcare financing sources [25]. The findings showed the income shares of per capita household expenditures and various financing sources across income quintiles. The authors revealed that 53.42% of total OOP expenditures were paid by the richest quintile (Q5). The KIs for Urban Employee Basic Medical Insurance (UEBMI) and OOP payments were statistically significantly positive, indicating that the wealthy contributed a larger proportion of healthcare payments compared to their income. In summary, the study found that UEBMI and OOP payments affect the overall KI of 0.0444, indicating that the healthcare financing system in North Jiangsu was progressive.

Another regional study conducted by Qin et al. (2017) in rural Guangxi Zhuang autonomous region examined the equity of healthcare financing based on two cross-sectional surveys that were carried out in 2009 and 2013 [26]. In this regional study, the situation regarding OOP was found to be regressive. Between 2009 and 2013, there was an observed rise in the relative share of tax and OOP payments among the low and middle consumption quintiles, accompanied by a decline in the high consumption quintiles. These findings indicate a discernible shift towards increased reliance on taxation and OOP expenditures for individuals with lower and moderate consumption levels, while individuals with higher consumption experienced a reduction in their tax and OOP burdens. This shift also contributed to a decrease in the Kakwani index from −0.1019 in 2009 to −0.1724 in 2013, reflecting the impact of these changes on income inequality. It is worth noting that this study identified a decrease in the number of households experiencing catastrophic health expenditures and attributed it to the effectiveness of the regional New Cooperative Medical Scheme (NCMS) model. Between 2009 and 2013, there was a significant reduction in the incidence of CHE, as measured by either total household expenditure (decreasing from 7.3% to 1.2%) or non-food expenditure (decreasing from 26.1% to 7.5%), indicating an improvement in households’ ability to afford healthcare expenses.

In the selected study of Zhou et al. (2022), the authors aimed to assess the progressivity of the merged insurance programs, namely the Urban Resident Basic Medical Insurance (URBMI) and the New Rural Cooperative Medical Scheme (NRCMS), by separately calculating progressivity measures for rural and urban areas [27]. After the consolidation of URBMI and NRCMS into Urban and Rural Resident Basic Medical Insurance (URRBMI), their findings indicate that OOP payments exhibited a near-proportional pattern in urban areas while displaying regressiveness in rural areas. In contrast to other modes of healthcare payment, OOP payments in the rural context operated as a post-paid health financing mechanism when healthcare providers may have a financial incentive to utilize costly drugs or advanced technologies under fee-for-service payment models. The overall equity in healthcare financing was lower in rural areas compared to urban areas. The observed regressivity primarily stemmed from a greater reliance on regressive financing methods in rural regions, including fixed URRBMI contributions, OOP expenses, and private health insurance premiums.

In South Korea, the high-income country of this region, Tae-Jin Lee et al. (2021) conducted a study to assess the vertical equity of healthcare financing over a considerable timeframe between 1990 and 2016 [28]. Their findings revealed that direct tax was the most progressive mode of healthcare financing, while OOP payments were consistently weakly regressive. While Korean medical insurance is characterized by its widespread coverage and a unified payer system, significant concerns have arisen regarding issues such as the equitable calculation of insurance premiums and the burden of high OOP expenses. In addition to copayments within the National Health Insurance, certain healthcare services are not covered, necessitating OOP payments from patients. Consequently, OOP payments accounted for roughly 36% of the total medical expenses in 2018. The trend of increased regressivity became more pronounced after 2006, partly attributable to the limited extent of health insurance coverage, which remained at approximately 63% during this period. Essentially, as healthcare expenditure continues to rise while insurance coverage remains stagnant, households experience a rise in their actual healthcare expenditure. Consequently, the increased regressivity suggests that individuals with lower incomes bear a disproportionate financial burden beyond their means.

It can be inferred that research conducted in East Asia acknowledges the significance of health insurance coverage, equitable burden-sharing within the formal sector among the working population, and the issue of differentials in payout sizes for the self-employed. Additionally, regional disparities in OOP payment inequality have been identified based on studies conducted in China.

### 3.4. Western Asia

The review examined multiple articles that provided insights into the distribution of healthcare expenses, OOP payments, and the impact of health sector reforms on financial protection in West Asia, focusing on Iran and Turkey. 

The first selected study provided in Iran by Jalali et al. revealed that the healthcare financing system in Shiraz, Iran, was regressive and that there was vertical inequity in healthcare OOP payments in 2018 [29]. This means that the low-income households paid a higher proportion of their income for health expenses than the high-income ones. The authors discussed the possible factors that contributed to this situation, such as the lack of universal insurance coverage, the high OOP payments, the low public financing, and the unequal distribution of health services. They also compared their findings with other studies conducted in Iran and other countries and highlighted the limitations of their study. They suggested some policy recommendations to improve equity in healthcare financing, such as expanding insurance coverage, redistributing income in the health sector to support low-income groups with the targeted subsidy plan, strengthening health insurance schemes, modifying health insurance benefit packages, and developing pro-poor strategies.

The next study from Iran, conducted by Rezaei et al., found that OOP payments for healthcare are inequitable, leading to a greater financial burden [30]. The study revealed an increase in the incidence of CHE over time (5.26% in 2017 for 40% thresholds), indicating that more households are spending a significant portion of their income on healthcare. To measure the progressivity of OOP payments, the authors utilized the Kakwani progressivity index and discovered that it has increased over time. This indicates that the healthcare financing mechanism through OOP payments has become fairer over the observation period. However, the Kakwani index values remain negative in the final period of observation, emphasizing the inequality experienced by poor households. The authors discussed the implications of these findings for healthcare policy in Iran and suggested further research to explore the factors contributing to the rise in OOP payments and CHE.

Abdi et al. conducted a study evaluating the impact of the Health Transformation Plan (HTP) in Iran on health spending [21]. The HTP aimed to reduce OOP expenses for inpatient care in public hospitals but led to increased payments for outpatient fees, ancillary services, and dental care. The implementation of the HTP resulted in a significant reduction in CHE (2.1% in 2015) and improved financial protection for all population groups. The Kakwani index indicated a slight increase in the progressivity of OOP health financing. Overall, the HTP had positive effects on health insurance coverage, reduced OOP payments, and a more equitable distribution of healthcare financing.

A Turkish study indicates that the financial burden of OOP health expenditures in Turkey still falls on individuals with low incomes, despite more than a decade of health reform. The progressive OOP health expenditure mechanism observed in the early years of reform shifted to a regressive one after comprehensive insurance policies were implemented. This picture, coupled with poor economic growth, has led to increased healthcare spending for individuals. The changing disease landscape from infectious diseases to chronic illnesses has also contributed to the rising costs of healthcare. While Turkey has made efforts to improve accessibility to healthcare services, there is a need to address disparities in insurance coverage, accessibility, and utilization between rural and urban areas [31].

Interestingly, two national studies from Iran yielded contrasting results, which could be attributed to differences in sample sizes. Studies in both Iran and Turkey emphasize the importance of healthcare system reforms aimed at reducing OOP payments for the population [30,31].

### 3.5. Europe, America, and Africa

We grouped these regions together because they are represented by a single type of country, and what unites all of them is their classification by the International Monetary Fund as countries with a high level of economic development.

The article by Citoni et al. analyzes the progressivity of healthcare financing in Italy at the regional level [32]. It finds that the Italian system is regressive overall, and more so in the southern regions than in the northern ones. It also shows that the interregional redistribution of value-added tax (VAT) revenues reduces but does not eliminate the regressivity of the system. The article discusses the implications of these findings for equity and policy, especially in light of the COVID-19 pandemic and its economic consequences. This heterogeneity arises from differences in tax rates and citizens’ ability to make contributions. The results also indicated that the health financing system in Italy, initially progressive, has become regressive due to the shift from direct to indirect taxation (i.e., this means a reduction in the share of corporate tax and personal income tax in favor of VAT) as the primary source of public funding. The authors call for greater investment in public healthcare with a higher weight given to progressive sources of financing to avoid further reduction in vertical equity. They also highlight the importance of monitoring vertical equity at both national and regional levels, as the post-COVID-19 pandemic crisis may exacerbate the relative disadvantage of southern regions and increase the share of the most regressive source of financing (VAT).

The study conducted in Portugal, despite indicating a reduction in inequality in out-of-pocket healthcare payments, underscores that it still remains high compared to the European region. A significant contributor to this inequality is the Kakwani index value for out-of-pocket payments for medications (−0.225). The authors note that following the 2008 economic crisis, the government implemented measures such as pension and salary reductions, lowering medication prices, shortening unemployment benefit periods, increasing income tax, property tax, excise duties, and VAT. While these cuts in benefits and tax increases primarily affected the affluent segments of the population, the study authors refrain from specifying whether these measures had a positive or negative impact on inequality in out-of-pocket healthcare payments [33].

Sterling Edmonds and Mohammad Hajizadeh investigated the progressivity and catastrophic effects of OOP expenditures for healthcare in Canada from 2010 to 2015 [34]. The findings indicated that OOP expenditures in Canada exhibited regressivity throughout the study period, as evidenced by negative Kakwani progressivity index values. A time-series regression analysis revealed a statistically significant increase in the regressivity of OOP expenditures from 2010 to 2015. The study also examined the proportion of households reaching the catastrophic threshold of 10% of total household consumption, indicating that 7% of Canadian households were affected by catastrophic OOP over the study period. Moreover, the study identified variations in the proportion of households affected by CHE across provinces and between urban and rural regions. It highlighted the burden of pharmaceutical drugs and dental services as major contributors to OOP healthcare expenditures, with rural households facing a higher proportion of pharmaceutical drug expenses compared to urban households. The results obtained at the regional level indicate that provinces with a higher proportion of rural population have a larger share of the population experiencing CHE and higher levels of inequality. Based on the Canadian experience, it appears that publicly funded health insurance solely for “medically necessary” healthcare is insufficient to ensure equitable healthcare financing, as OOP expenditures not covered by insurance can impose catastrophic and inequitable financial burdens on individuals.

According to a study conducted by Nundoochan in Mauritius, OOP expenditures remain relatively high and significant among the poorest segments of the population, particularly in the absence of social health insurance [35]. This situation is further exacerbated by a widespread perception of low service quality in the public sector. Consequently, there is an increased risk of CHE and the impoverishment of the poorest households. The author discusses that a considerable portion of OOP spending is attributed to the purchase of pharmaceuticals from the private sector. This trend may be influenced by a widespread misconception that generic drugs available in the public sector are of inferior quality. The author has noted that the COVID-19 pandemic has revealed gaps in the implementation of interventions related to UHC, which are designed to ensure access to a comprehensive range of healthcare services, including health promotion, prevention, and treatment. The emphasis on healthcare investments has been skewed towards curative care, compared to more cost-effective approaches such as health promotion and disease prevention. It is critically important to strike a balance between advocacy, prevention, and treatment in the pursuit of equitable distribution of healthcare services to achieve UHC.

Studies from three countries in Europe, North America, and Africa emphasize the importance of subnational investigation of inequality, high expenditures on pharmaceuticals, and the impact of the COVID-19 pandemic on both OOP healthcare spending and overall healthcare financing systems.

## 4. Discussion

The objective of this systematic review was to assess OOP health expenditure inequality measured by the FIA across different territories, in the context of achieving UHC by 2030. All studies examining equity in OOP expenditures for healthcare were included in this review, utilizing FIA and the Kakwani index as assessment tools. Some of the studies included in the review contained data on CHE, which served as an additional indicator of excessive financial burden due to OOP costs. The findings of this review are presented independently. 

Quite contradictory results were obtained in the countries of South and Southeast Asia. Among all the countries reviewed, Malaysia is one of three countries with a progressive OOP payment system, while Bangladesh and India have regressive systems. A recent study from Bangladesh indicates an escalation in the Kakwani index, making it the most negative index among all the publications examined [20]. 

However, it is worth noting that in O‘Donnell et al.’s study, Bangladesh is characterized as a country with positive progressivity in the overall financing system, including direct and indirect taxes as well as direct payments. Additionally, the study highlights that the healthcare system is predominantly OOP-financed. In India, a high proportion of healthcare costs are paid OOP [36]. Out-of-pocket (OOP) health expenditures increased by 62% from 2004 to 2014, while CHE increased by 17% over the same period [37]. This suggests that the financial protection offered to patients by the healthcare system in India remains inadequate, especially in rural areas with a high proportion of CHE and a regressive OOP payment system [24]. 

In Malaysia, public funding is the primary source of health financing, accounting for 51% of total health spending in 2018. Out-of-pocket (OOP) health expenditures were 35% of total health spending [38]. The Malaysian health financing system has not changed significantly since 1998/99, when it was found to be progressive, with a Kakwani index of 0.217 for total health expenditures. Out-of-pocket (OOP) health expenditures were also mildly progressive in the same period, with a Kakwani index of 0.010. The national (macro) level expenditure showed that the government subsidized 58.2% of the funding in the public health sector [23,39].

The study demonstrating progressivity in OOP payments was conducted in the province of North Jiangsu. These findings align with previous results, which indicate a significant prevalence of both social and private insurance alongside progressivity [25,36]. In Guangxi, an autonomous region, a reverse trend was observed, indicating a regression in out-of-pocket expenditures among the population with low and moderate consumption levels. This was significantly influenced by the regional disparity in the NCMS, where more economically developed regions and population groups benefit to a greater extent [26,40]. In another selected study in China, more pronounced inequality in OOP expenditures was found among rural populations, confirming previous findings that rural residents from lower-income groups face high financial risk and should be a priority target for future reforms in achieving UHC [27,41].

Theoretically, positive Kakwani index values correspond to a fair financing system. However, caution should be exercised regarding positive values for out-of-pocket healthcare payments, as they may indicate lower access to healthcare services. This paradoxical situation can be explained, firstly, by the fact that in some countries, the poor cannot afford to spend money on healthcare and primarily rely on free government services. Secondly, individuals with high incomes can afford more paid services that require out-of-pocket payments [13,42].

In another country within the region, South Korea, a developed high-income country, a survey-based study based on household data from 1990 to 2016 emphasizes the significance of the healthcare insurance system and its impact on fair distribution of financial burden [28]. The study revealed a weak regressivity in OOP expenditures, which may not affect individuals with low incomes in terms of access to doctors or outpatient services. However, CHEs in the poorest group were approximately 20 times higher than those in the richest group, which could impact access to advanced medical services. This may also be associated with the fact that the share of OOP expenditures in total healthcare spending in South Korea is approximately 1.5 times higher than that in OECD countries [43].

The third study from the Western Asia region, specifically from Iran, published in the *Eastern Mediterranean Health Journal*, indicates that after the implementation of the HTP, OOP payments for healthcare became progressive. The study also highlights that the HTP reduced the proportion of households experiencing CHE [21]. However, two other studies from Iran, conducted at both regional and national levels, demonstrate the regressivity of OOP payments and an increase in the proportion of the population facing CHE. All studies emphasize the high share of OOP payments, reaching nearly 50%, in the healthcare financing system [29,30]. The HTP, implemented in 2014, effectively tackled challenges such as high OOP payments and brought significant improvements to the healthcare system. It successfully balanced the health budget, provided risk protection for the entire population, and enhanced access, quality, equity, and satisfaction in healthcare services [44]. Another country in the region, but with a much more regressive system than Iran, is Turkey. Moreover, the strong trend of regressivity persisted for over 10 years, despite the reduction in the proportion of self-employed individuals and the integration of subsidies into a unified social insurance system [31]. These findings are supported by data indicating that the wealthy segments of the population benefited more from the reforms, while the increase in OOP expenditures may be linked to informal payments, which is a significant problem in Turkey [45]. 

In addition to the study in South Korea, our review includes studies from high-income countries such as Italy and Canada. The results of the Canadian study indicate a slight regressivity in OOP payments within a tax-based public healthcare system [34]. Overall, the financing system is relatively proportional, but the shift to indirect taxes, the lack of government-provided ophthalmological and dental care, and a high proportion of OOP payments for outpatient prescribed medications contribute to an increasing burden among the poorest segments of the population [46,47]. Italy, a country with a tax-based national healthcare financing system, faces challenges in healthcare funding following the global financial crisis. The responsibility for administering and managing the system lies with the regions [48]. Observations indicate that the OOP payment system in Italy is regressive overall, particularly in the southern regions. Although interregional redistribution of tax revenues partially mitigates this regressivity, it does not fully eliminate it. In the context of the economic crisis following the COVID-19 pandemic, coupled with the geographic concentration of privately insured individuals in wealthier northern regions, the relative disadvantage of low-income individuals from southern regions may increase due to rising OOP payments. The study authors emphasize the need to increase the use of progressive healthcare financing mechanisms, especially in the context of the COVID-19 pandemic and its economic consequences [32]. Mauritius, a high-income country, had a tax-based public healthcare system that covered 70% of services. However, it has recently experienced an increase in the share of OOP payments in healthcare financing, a rise in households facing CHE, and an increased likelihood of economic downturn due to the COVID-19 pandemic. Despite these challenges, the overall healthcare financing system in Mauritius is relatively proportional, with progressive direct taxes and voluntary insurance. The regressivity of OOP payments is attributed by the authors to patients’ irrational preference for purchasing patented drugs instead of free generics [35,49]. 

Based on the analyzed research, achieving UHC remains a challenge due to regional disparities; disparities between urban and rural areas; income inequality; the shift of the burden from infectious diseases to chronic illnesses; the imbalance between advocacy, prevention, and treatment; and the additional burden observed during the COVID-19 pandemic. It has been found that individuals who have contracted COVID-19 face increased healthcare expenses post-recovery [50,51,52]. Furthermore, there is a likelihood of increased utilization of healthcare services post-COVID-19, driven by pent-up demand for medical services following periods of lockdowns. This underscores the need for further examination of the inequality associated with the global healthcare crisis of the COVID-19 pandemic [53,54,55].

A limitation of this study was the lack of access to all scientific databases. The systematic review included a limited number of countries due to the scarcity of publications that aligned with the specified objectives and methodologies. There were some gaps and inconsistencies in the data presented across different studies, which constrained our review. Not all studies were based on national samples, and some relied on distinctive sets of primary data. These limitations may explain the contradictory results observed in some countries. Another limitation arises from the difference in thresholds used to measure CHE. Moreover, not all studies included in this review presented data on CHE.

## 5. Conclusions

The results of this systematic review indicate that healthcare financing systems that are heavily reliant on direct taxation and higher public funding relative to OOP payments tend to have a higher proportion of the population experiencing catastrophic health expenditure (CHE) and exhibit a more regressive healthcare system. Despite progress, achieving UHC remains a challenging task for governments, as financial inequality persists among vulnerable populations. Challenges such as high OOP payments for pharmaceutical goods, formal charges, and regional disparities in healthcare financing administration contribute to this complexity. The impact of the COVID-19 pandemic adds another layer of complexity to these challenges, necessitating further investigation into its effects on healthcare financing systems and their equitable distribution. Both high- and low- to middle-income countries are advised to reduce OOP healthcare financing mechanisms as they contribute to increasing inequality between the rich and poor segments of the population. Effective measures to reduce inequality in low- to middle-income countries include reducing the proportion of self-employed individuals not covered by social or mandatory insurance systems. For high-income countries, it is necessary to monitor payment or copayment systems for medications and implement fair taxation between affluent and disadvantaged regions and populations. This is particularly crucial due to the crisis resulting from the COVID-19 pandemic. Continued monitoring and healthcare system reforms are crucial to address these issues and work towards more equitable and accessible healthcare for all.

## Figures and Tables

**Figure 1 healthcare-12-01051-f001:**
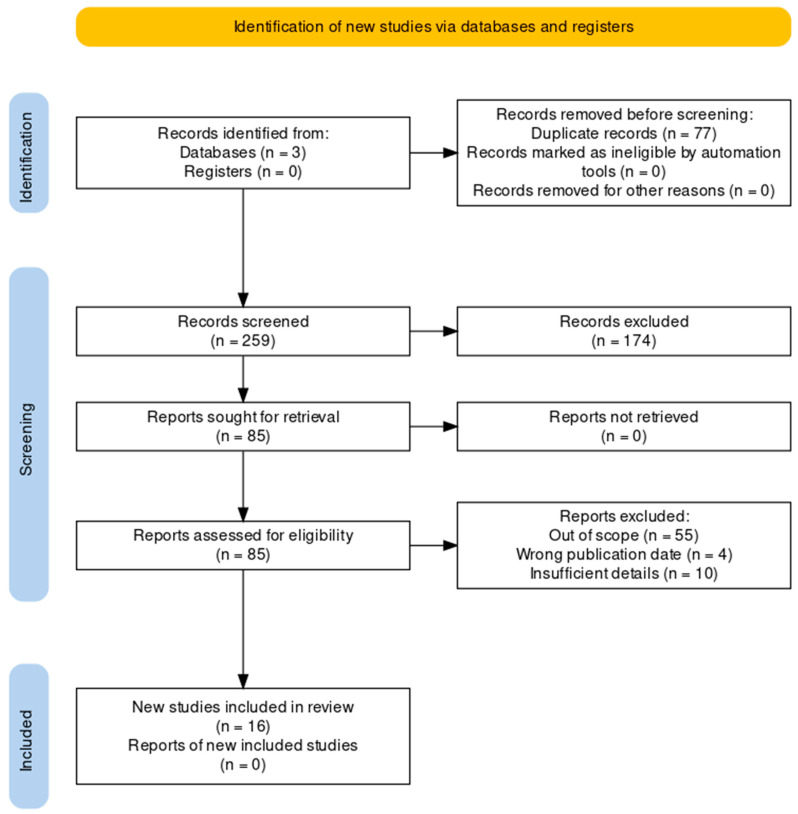
PRISMA flowchart.

**Figure 2 healthcare-12-01051-f002:**
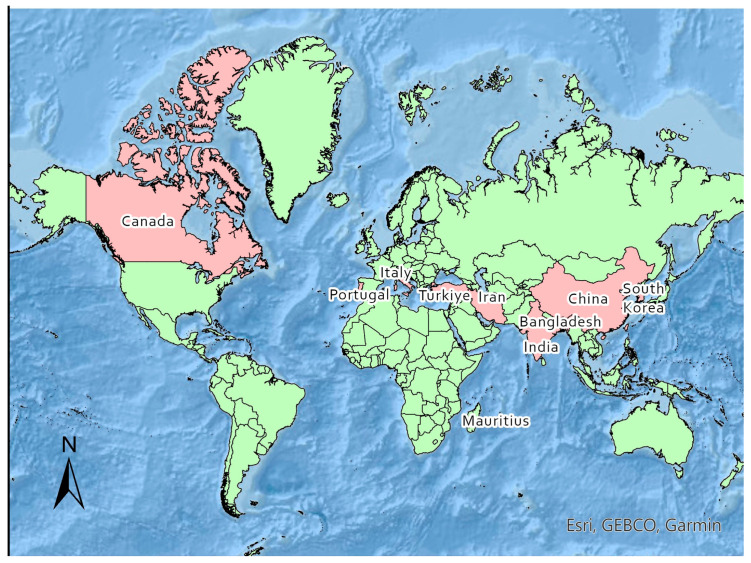
Map of study locations.

**Figure 3 healthcare-12-01051-f003:**
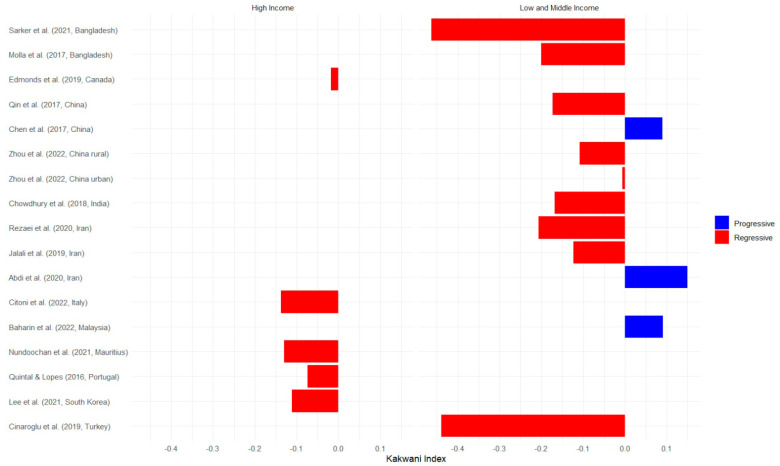
OOP Kakwani index values of selected studies by income country classification [20,21,22,23,24,25,26,27,28,29,30,31,32,33,34,35].

**Table 1 healthcare-12-01051-t001:** Attributes of the studies encompassed within the review [20,21,22,23,24,25,26,27,28,29,30,31,32,33,34,35].

	Author (Year)	Country	Sample Size	Data Sources	Study Level	Equity Analysis	CHE Analysis
1	Molla et al. (2017) [22]	Bangladesh	12,240 households, 55,580 individuals	Bangladesh Household Income and Expenditure Survey	National	OOP KI *, other type of spending KI *	-
2	Sarker et al. (2021) [20]	Bangladesh	46,075 households 186,207 individuals	Household Income and Expenditure Survey	National	OOP KI	-
3	Edmonds et al. (2019) [34]	Canada	33,367 individuals	Canada’s Survey of Household Spending	Subnational (10 regions)	OOP KI	10% of total current household consumption
4	Qin et al. (2017) [26]	China	4634 households in 2009, 3951 in 2013	Mixed: Survey and China Taxation Development Report	Regional, 2 districts from Guangxi Province	OOP KI, other type of spending KI	40% of the household’s capacity to pay or non-subsistence spending
5	Chen et al. (2017) [25]	China	3008 households, 8854 individuals	Survey	Regional North Jiangsu Province	OOP KI, other type of spending KI	-
6	Zhou et al. (2022) [27]	China	6000 households, 6527 individuals	Survey	Regional, Heilongjiang Province	OOP KI, other type of spending KI	-
7	Chowdhury et al. (2018) [24]	India	12,134 households, 62,335 individuals	Survey	Subnational (3 states)	OOP KI	20% of total current household consumption
8	Jalali et al. (2019) [29]	Iran	740 households 2357 individuals	Survey	Regional (Shiraz is the capital of Fars Province)	OOP KI, other type of spending KI	-
9	Abdi et al. (2020) [21]	Iran	9535 households in 2014, 9543 in 2015	Household Expenditure and Income Survey	National	OOP KI	40% of the household’s capacity to pay or non-subsistence spending
10	Rezaei et al. (2020) [30]	Iran	18,582 households in 1991, 21,854 in 1996, 26,714 in 2001, 31,111 in 2011, 38,220 in 2014, 37,860 in 2017	Household income and expenditure survey (HIES) of Iran	National	OOP KI	40, 30, 20% of the household’s capacity to pay or non-subsistence spending
11	Citoni et al. (2022) [32]	Italy	15,013 households	Italian National Institute of Statistics (ISTAT) Household Budget Survey	National	OOP KI, other type of spending KI	-
12	Baharin et al. (2022) [23]	Malaysia	14,437 households	Household Expenditure Survey (HES)	National	OOP KI	-
13	Nundoochan et al. (2021) [35]	Mauritius	2700 households 8870 individuals	Survey	National	OOP KI, other type of spending KI	-
14	Quintal and Lopes (2016) [33]	Portugal	9489 households	Portuguese Household Budget Survey	National	OOP KI	40% of the household’s capacity to pay or non-subsistence spending
15	Lee et al. (2021) [28]	South Korea	Ranging between 5500 and 7500 households between 1990 and 2016	Household Income and Expenditure Survey (HIES)	National	OOP KI, other type of spending KI	-
16	Cinaroglu et al. (2019) [31]	Turkey	Ranging between 25,764 and 11,491 households between 2003 and 2015	Turkish Household Budget Survey	National	OOP KI	-

* KI—Kakwani index.

**Table 2 healthcare-12-01051-t002:** Kakwani index values of all financing sources.

Author	Country	Total	OOP	Total Taxation	Direct Tax	Indirect Tax	Social Insurance	Private Insurance	CHE
Molla et al. (2017) [22]	Bangladesh	−0.1917	−0.2005	−0.0714	-	-	−0.2094	−0.4342	-
Sarker et al. (2021) [20]	Bangladesh	-	−0.463	-	-	-	-	-	-
Edmonds et al. (2019) [34]	Canada	-	−0.0174	-	-	-	-	-	7%
Qin et al. (2017) [26]	China	−0.1636	−0.1724	0.0013	-	-	−0.2865	-	7.5%
Chen et al. (2017) [25]	China	0.0444	0.0896	−0.0241	-	-	-	-	-
Zhou et al. (2022) [27]	China urban	−0.0142	−0.0064	-	0.4628	0.0009	-	−0.0104	-
	China rural	−0.1208	−0.1078	-	0.4087	0.0284	-	−0.0842	-
Chowdhury et al. (2018) [24]	India	-	−0.168	-	-	-	-	-	9.4% urban 19.2% rural
Jalali et al. (2019) [29]	Iran	−0.112	−0.123	-	-	−0.038	−0.125	-	-
Abdi et al. (2020) [21]	Iran	-	0.15	-	-	-	-	-	2.1%
Rezaei et al. (2020) [30]	Iran	-	−0.207	-	-	-	-	-	5.26%
Citoni et al. (2022) [32]	Italy	−0.099	−0.137	-	-	-	-	0.017	-
Baharin et al. (2022) [23]	Malaysia	-	0.0910	-	-	-	-	-	-
Nundoochan et al. (2021) [35]	Mauritius	−0.004	−0.13	-	0.30	-	-	0.10	-
Quintal and Lopes (2016) [33]	Portugal	-	−0.074	-	-	-	-	-	2.1%
Lee et al. (2021) [28]	South Korea	−0.014	−0.111	-	0.330	−0.030	0.023	−0.050	-
Cinaroglu et al. (2019) [31]	Turkey	-	−0.44	-	-	-	-	-	-

## Data Availability

Data are available on request from the authors.

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
