# Peer review of "Inequalities in Out-of-Pocket Health Expenditure Measured Using Financing Incidence Analysis (FIA): A Systematic Review"

_healthcare, 2024, doi:10.3390/healthcare12101051_

Round 1

Reviewer 1 Report

Comments and Suggestions for Authors

Studying (in)equity in accessing healthcare is certainly not new in the literature. Health care differences between countries and population groups within countries are due to many economic, institutional, and cultural factors. They include differences in economic development, level of economic freedom, healthcare systems, insurance schemes, the types of healthcare (e.g., primary and secondary), quality of services, foreign financial assistance, fiscal centralization or decentralization, presence of solidarity economy, etc.  To ensure that the systematic review study captures these differences, it should aim to cover as many countries as possible.

The stated aim of this systematic review was "to assess changes in inequalities in out-of-pocket health expenditure, as measured by the financing incidence analysis, across territories in the context of achieving universal health coverage (UHC) by 2030” (85-87), and “The objective of this systematic review was to evaluate the evidence regarding equity in healthcare financing mechanisms” (446-447). The aims/objectives formulated in the two parts of the work are not entirely consistent. I would like to point out that systematic reviews aim to gather all the evidence on a specific research question and involve a reproducible and thorough search of the literature and critical appraisal of the eligible studies. So, what are the research questions of the study?

1.      Abstract: I suggest excluding the terms Background, Methods, Results, and Conclusions from the abstract.

2. The front matter of the manuscript does not follow the remaining structure (please compare the first page with the Healthcare Microsoft Word template).

3.      Introduction:

·        45 - When mentioning the first  Sustainable Development Goals, it is worth pointing out which goal focuses specifically on health protection. Goal 3: Ensure healthy lives and promote well-being for all at all ages

4.      Authors’ literature searches included two American databases (PubMed and Web of Science) and one European database (Scopus) for papers published from 2016 to 2022.

5. A systematic review of the literature has finally included 15 studies. Was it a target number? Did the authors modify their search strategies or inclusion/exclusion  criteria to achieve the desired number of papers? Did the authors check the bibliographies of all relevant literature in the review for further relevant bibliographies? The literature review is quantitatively biased towards developing countries, as most of the selected studies deal with them. It ignores several developed countries, e.g., no entries for the USA, Australia, Israel…

6.      2.3. Data Extraction: I think it is more appropriate to use the full names of the researchers who contributed to the study rather than their initials.

7.      Based on the results reported in Figure 3 and Table 2, it has been observed that the out-of-pocket (OOP) healthcare expenditure can be both regressive (Kakwani index with a negative value) and progressive (Kakwani index with a positive value) in the same country, for instance, China and Iran. This raises the question of what causes such a contradiction in the results obtained for the same country. Moreover, in which situations will out-of-pocket payments be progressive? Which is better (more fair), regressive or progressive, for developing and developed countries? It would be interesting to see the results after dividing the literature according to the two study country groups: High-income and Low- and Middle-Income countries.

8.      Discussion:

·        540 – „Limitation of this study was the lack of access to all scientific databases. The systematic review included a limited number of countries due to the scarcity of publications that aligned with the specified objectives and methodologies”. Did the authors attempt to search for literature in Google Scholar or ProQuest? I made such an attempt searching in Google Scholar, selecting the same period (2016-2022) and adding country names as keywords. I should not suggest specific references, so I leave it to the authors to include other countries or to rephrase the justification for including a limited number of countries in the systematic review.

·        How are the results presented in this article aligned with those from systematic review studies in earlier periods?

9.      Conclusions: Any Authors' reflection on the positive and negative consequences of implementing out-of-pocket (OOP) payments as a source of healthcare financing? Are there any concrete recommendations for health policy, except for the general statement that continued monitoring and healthcare system reforms are crucial?

10.   References:

·        Errors in bibliography: “Agyepong IA, Adjei S. Public social policy development and implementation: a case study of the Ghana National Health Insurance  scheme. Health Policy Plan. 2007;23:150–60”. After checking: Irene Akua Agyepong, Sam Adjei, Public social policy development and implementation: a case study of the Ghana National Health Insurance scheme, Health Policy and Planning, Volume 23, Issue 2, March 2008, Pages 150–160, https://doi.org/10.1093/heapol/czn002 .

·        A request to carefully re-check the correctness of all bibliographic records of the literature items.

·        Healthcare editors encourage the use of DOI

 Here are the comments by lines of the article

50 – “Barber et al. identified four challenges for financing the WHO's goals and primary health care.” – The four challenges should be presented one after the other in numerical order.

61 – “In Burkina Faso, interventions related to UHC reduced inequalities in healthcare spending distribution and improved access to healthcare services, especially for the poor [5]”. Reference item #5 relates to Ghana and presents results for Ghana, not Burkina Faso. Having read this article on Ghana, I identified a contradiction between the quoted statement and the original information. One of the fragments: “However, these achievements were accompanied by inequities in financial access to basic and essential clinical services” (p. 154).   Additionally, there is a mistake in the year of publication of the article [5].

82 – “there are several systematic reviews on the topic [12, 13]” – It seems that citing only two literature items to show previous systematic reviews is insufficient, especially that they refer only to Low- and Middle-Income Countries.

141 – “UHC refers to a system where all individuals and communities can access quality healthcare services” (should be “access to quality”)

195 - Table 1 – lines 4, 9, 10 –  there is no need to repeat "households" if all samples apply to them.

466 - I suggest splitting the paragraph from line 466 into two parts (Malaysia, China) with a clear indication that the provinces being described are in China.

540 – „Limitation of this study was the lack of access to all scientific databases. The systematic review included a limited number of countries due to the scarcity of publications that aligned with the specified objectives and methodologies”. Did the authors attempt to search for items in Google Scholar or Proquets?

577 - Is a list of abbreviations at the end of the article required?

Comments on the Quality of English Language

No comments

Author Response

Response to Reviewers:

Journal: Healthcare

Manuscript No: healthcare-2991471

Manuscript title: Inequalities in out-of-pocket health expenditure measured using financing incidence analysis (FIA): A systematic review.

Corresponding author: Askhat Shaltynov

Referee 1

Changes by the authors

The stated aim of this systematic review was "to assess changes in inequalities in out-of-pocket health expenditure, as measured by the financing incidence analysis, across territories in the context of achieving universal health coverage (UHC) by 2030” (85-87), and “The objective of this systematic review was to evaluate the evidence regarding equity in healthcare financing mechanisms” (446-447). The aims/objectives formulated in the two parts of the work are not entirely consistent. I would like to point out that systematic reviews aim to gather all the evidence on a specific research question and involve a reproducible and thorough search of the literature and critical appraisal of the eligible studies. So, what are the research questions of the study?

Thank you very much for taking the time to review this manuscript. Thank you for your thoughtful comments. We have addressed all the proposed amendments and highlighted them in yellow.

We have harmonized the research aim across both sections of the manuscript. The aim of this systematic review was to assess OOP health expenditures inequality measured by the FIA across different territories, in the context of achieving UHC by 2030.

Abstract: I suggest excluding the terms Background, Methods, Results, and Conclusions from the abstract.

Thank you for your suggestion.

We excluded the terms "Background," "Methods," "Results," and "Conclusions" from the abstract.

The front matter of the manuscript does not follow the remaining structure (please compare the first page with the Healthcare Microsoft Word template).

We adjusted the front matter of the manuscript to align with the remaining structure, following the Healthcare Microsoft Word template. We transferred registration information to the end of Methods section: 2.8. Registration

The systematic review protocol in the PROSPERO database ID: CRD42023395936.

45 - When mentioning the first  Sustainable Development Goals, it is worth pointing out which goal focuses specifically on health protection. Goal 3: Ensure healthy lives and promote well-being for all at all ages

We revised the text to specifically denote SDG Target 3.8, which aims to achieve universal health coverage (UHC). Universal health coverage (UHC) is a key mechanism to provide financial protection in healthcare, and it has been recognized as a key target in the Sustainable Development Goal (SDG) 3.8 adopted by the United Nations in 2015

Authors’ literature searches included two American databases (PubMed and Web of Science) and one European database (Scopus) for papers published from 2016 to 2022.

5. A systematic review of the literature has finally included 15 studies. Was it a target number? Did the authors modify their search strategies or inclusion/exclusion  criteria to achieve the desired number of papers? Did the authors check the bibliographies of all relevant literature in the review for further relevant bibliographies? The literature review is quantitatively biased towards developing countries, as most of the selected studies deal with them. It ignores several developed countries, e.g., no entries for the USA, Australia, Israel…

We did not set a specific target value for the number of publications included in the review. The search strategy, inclusion and exclusion criteria were designed to capture all existing evidence in the databases using Boolean operators, synonyms, and alternative terms in the search query.

The development of inclusion and exclusion criteria took into account all parameters relevant to the research objective, namely the examination of vertical inequality/progressivity/FIA in out-of-pocket healthcare payments measured by the Kakwani index. The period for the selected studies was chosen considering previously published reviews on the topic. Indeed, during this period, studies employing such methods predominated in low- and middle-income countries. Studies in high-income countries were mainly conducted before 2016 or gained popularity after 2022. Examples of studies are provided as references:

Ketsche, Patricia; Adams, E. Kathleen; Wallace, Sally; and Kannan, Viji, "The Distribution of the Burden of

US Health Care Financing" (2015). GHPC Articles. 161.

https://scholarworks.gsu.edu/ghpc_articles/161

Nübler L, Busse R, Siegel M. The role of consumer choice in out-of-pocket spending on health. Int J Equity Health. 2023 Jan 31;22(1):24. doi: 10.1186/s12939-023-01838-1. PMID: 36721164; PMCID: PMC9890873.

Law HD, Marasinghe D, Butler D, Welsh J, Lancsar E, Banks E, Biddle N, Korda R. Progressivity of out-of-pocket costs under Australia's universal health care system: A national linked data study. Health Policy. 2023 Jan;127:44-50. doi: 10.1016/j.healthpol.2022.10.010. Epub 2022 Oct 21. PMID: 36456400.

Shmueli, A., Achdut, L., & Sabag-Endeweld, M. (2008). Financing the package of services during the first decade of the national health insurance law in Israel: Trends and issues. Health Policy, 87(3), 273–284. doi:10.1016/j.healthpol.2008.02.008

2.3. Data Extraction: I think it is more appropriate to use the full names of the researchers who contributed to the study rather than their initials.

We updated the sections 2.3 and 2.4 by replacing the initials with the full names of the researchers who contributed to the study.

Based on the results reported in Figure 3 and Table 2, it has been observed that the out-of-pocket (OOP) healthcare expenditure can be both regressive (Kakwani index with a negative value) and progressive (Kakwani index with a positive value) in the same country, for instance, China and Iran. This raises the question of what causes such a contradiction in the results obtained for the same country. Moreover, in which situations will out-of-pocket payments be progressive? Which is better (more fair), regressive or progressive, for developing and developed countries? It would be interesting to see the results after dividing the literature according to the two study country groups: High-income and Low- and Middle-Income countries.

Thank you for the comment. We have divided Figure 3 into two graphs: one for high-income countries and one for low- to middle-income countries. In the text, we emphasized studies from high-income countries:

In South Korea, the high-income country of this region, Tae-Jin Lee et al. (2021) conducted a study to assess the vertical equity of health care financing over a considerable timeframe between 1990 and 2016.

We grouped these regions together because they are represented by a single type of country, and what unites all of them is their classification by the International Monetary Fund as countries with a high level of economic development.

The regional structure of the text was designed to assess progress in reducing inequality compared to previous studies, highlighting regional disparities in inequality:

Asante A, Price J, Hayen A, Jan S, Wiseman V. Equity in Health Care Financing in Low- and Middle-Income Countries: A Systematic Review of Evidence from Studies Using Benefit and Financing Incidence Analyses. PLoS One. 2016 Apr 11;11(4):e0152866. doi: 10.1371/journal.pone.0152866. PMID: 27064991; PMCID: PMC4827871.

O'Donnell O, van Doorslaer E, Rannan-Eliya RP, et al. Who pays for health care in Asia?. J Health Econ. 2008;27(2):460-475. doi:10.1016/j.jhealeco.2007.08.005

Kolasa, K., Kowalczyk, M. Does cost sharing do more harm or more good? - a systematic literature review. BMC Public Health 16, 992 (2016). https://doi.org/10.1186/s12889-016-3624-6

We did not discuss differences in the indices within a single country, as Table 3 provides sources for the original studies. For China, these were different regions and sample sizes. For Iran, we added a sentence stating that the difference is due to variations in sample sizes:

Interestingly, two national studies from Iran yielded contrasting results, which could be attributed to differences in sample sizes.

For both developed and developing countries, a progressive system with positive Kakwani index values indicates a fairer payment system. However, developing countries may face a paradoxical situation with positive index values, as discussed in the discussion section:

Theoretically, positive Kakwani index values correspond to a fair financing system. However, caution should be exercised regarding positive values for out-of-pocket healthcare payments, as they may indicate lower access to healthcare services. This paradoxical situation can be explained, firstly, by the fact that in some countries, the poor cannot afford to spend money on healthcare and primarily rely on free government services. Secondly, individuals with high incomes can afford more paid services that require out-of-pocket payments [13,42].

„Limitation of this study was the lack of access to all scientific databases. The systematic review included a limited number of countries due to the scarcity of publications that aligned with the specified objectives and methodologies”. Did the authors attempt to search for literature in Google Scholar or ProQuest? I made such an attempt searching in Google Scholar, selecting the same period (2016-2022) and adding country names as keywords. I should not suggest specific references, so I leave it to the authors to include other countries or to rephrase the justification for including a limited number of countries in the systematic review. How are the results presented in this article aligned with those from systematic review studies in earlier periods?

Thank you for your valuable feedback. We conducted a search with similar search strategy on Google Scholar and ProQuest for a similar period of research and obtained 17200 results for Google Scholar and 5155 articles from ProQuest. After selecting publications and removing duplicates, we identified 2 studies: Mercier G, Pastor J, Clément V, Rodts U, Moffat C, Quéré I (2019) "Out-of-pocket payments, vertical equity and unmet medical needs in France: A national multicenter prospective study on lymphedema." PLoS ONE 14(5): e0216386. and

Quintal C, Lopes J. "Equity in health care financing in Portugal: findings from the Household Budget Survey 2010/2011." Health Economics, Policy and Law. 2016;11(3):233-252. Both studies were also found in the Scopus database. The study from France only assessed the Kakwani index for out-of-pocket healthcare expenses for lymphedema patients, while the study from Portugal was published online in 2015. Nevertheless, we decided to include the study from Portugal in our review as it appears in databases as a 2016 study. Accordingly, we updated Figures 1,2 and 3, Tables 1 and 2, and added a paragraph to the group of developed countries from Europe, America, and Africa:

The study conducted in Portugal, despite indicating a reduction in inequality in out-of-pocket healthcare payments, underscores that it still remains high compared to the European region. A significant contributor to this inequality is the Kakwani index value for out-of-pocket payments for medications (-0.225). The authors note that fol-lowing the 2008 economic crisis, the government implemented measures such as pen-sion and salary reductions, lowering medication prices, shortening unemployment benefit periods, increasing income tax, property tax, excise duties, and VAT. While these cuts in benefits and tax increases primarily affected the affluent segments of the population, the study authors refrain from specifying whether these measures had a positive or negative impact on inequality in out-of-pocket healthcare payments [33].

Our study results differ from previous ones in that they focus on assessing inequality specifically in OOP healthcare expenditures, factors influencing this inequality, and measures that can reduce it. Thus, in accordance with the valuable recommendations from the reviewers, we have included these measures in the conclusions of our review.

Conclusions: Any Authors' reflection on the positive and negative consequences of implementing out-of-pocket (OOP) payments as a source of healthcare financing? Are there any concrete recommendations for health policy, except for the general statement that continued monitoring and healthcare system reforms are crucial?

Thank you for the valuable feedback. We have added specific recommendations for both high-income and low- to middle-income countries. Both high- and low- to middle-income countries are advised to reduce OOP healthcare financing mechanisms as they contribute to increasing inequality between the rich and poor segments of the population. Effective measures to reduce inequality in low- to middle-income countries include reducing the proportion of self-employed individuals not covered by social or mandatory insurance systems. For high-income countries, it is necessary to monitor payment or co-payment systems for medications and implement fair taxation between affluent and disadvantaged regions and populations. This is particularly crucial due to the crisis resulting from the COVID-19 pandemic.

Errors in bibliography: “Agyepong IA, Adjei S. Public social policy development and implementation: a case study of the Ghana National Health Insurance  scheme. Health Policy Plan. 2007;23:150–60”. After checking: Irene Akua Agyepong, Sam Adjei, Public social policy development and implementation: a case study of the Ghana National Health Insurance scheme, Health Policy and Planning, Volume 23, Issue 2, March 2008, Pages 150–160, https://doi.org/10.1093/heapol/czn002 .

·        A request to carefully re-check the correctness of all bibliographic records of the literature items.

·        Healthcare editors encourage the use of DOI

We apologize for the technical error in using the Zotero reference manager. We re-checked the reference list.

50 – “Barber et al. identified four challenges for financing the WHO's goals and primary health care.” – The four challenges should be presented one after the other in numerical order.

Thank you for the feedback. The four challenges identified by Barber et al. have been incorporated into the text in numerical order: 1. Global normative expenditure targets were primarily devised for advocacy purposes, emphasizing the significance of healthcare for national development and securing political commitment; 2. Focusing attention on global normative targets may lead to the mistaken assumption that achieving Universal Health Coverage (UHC) is a fixed threshold or singular, unchanging goal over time; 3. The concept of a global normative target assumes that all countries need to spend a certain amount on healthcare to achieve similar outcomes; 4. Global normative targets typically focus attention solely on funding deficits, leading some policymakers and donors to assume that private financing could fill the gap left by limited government budgetary capacity.

61 – “In Burkina Faso, interventions related to UHC reduced inequalities in healthcare spending distribution and improved access to healthcare services, especially for the poor [5]”. Reference item #5 relates to Ghana and presents results for Ghana, not Burkina Faso. Having read this article on Ghana, I identified a contradiction between the quoted statement and the original information. One of the fragments: “However, these achievements were accompanied by inequities in financial access to basic and essential clinical services” (p. 154).   Additionally, there is a mistake in the year of publication of the article [5].

We apologize for the oversight. The correct source for this excerpt is the publication by De Allegri M, Rudasingwa M, Yeboah E, Bonnet E, Somé PA, Ridde V. Does the implementation of UHC reforms foster greater equality in health spending? Evidence from a benefit incidence analysis in Burkina Faso. BMJ Glob Health. 2021 Dec;6(12):e005810. doi: 10.1136/bmjgh-2021-005810. PMID: 34880059; PMCID: PMC8655516.

We changed the reference list.

82 – “there are several systematic reviews on the topic [12, 13]” – It seems that citing only two literature items to show previous systematic reviews is insufficient, especially that they refer only to Low- and Middle-Income Countries.

Thank you for your comment. You are correct. We referenced two systematic reviews encompassing studies from low- and middle-income countries because only these reviews included studies examining vertical inequality in out-of-pocket payments, assessed using the Kakwani progressivity index. We have changed the word "several" to "two" in the text, and we have also clarified the sentence to: "the impact of vertical inequality in out-of-pocket payments on healthcare." A range of international studies evaluated the impact of vertical inequality in OOP payments on healthcare, and there are two systematic reviews on the topic

141 – “UHC refers to a system where all individuals and communities can access quality healthcare services” (should be “access to quality”)

We have corrected our oversight. UHC refers to a system where all individuals and communities can access to quality healthcare services without suffering financial hardship.

195 - Table 1 – lines 4, 9, 10 –  there is no need to repeat "households" if all samples apply to them.

We have removed all repetitions of the word "household" in Table 1.

466 - I suggest splitting the paragraph from line 466 into two parts (Malaysia, China) with a clear indication that the provinces being described are in China.

Thank you for your suggestion. We have separated Malaysia and China into individual paragraphs.

540 – „Limitation of this study was the lack of access to all scientific databases. The systematic review included a limited number of countries due to the scarcity of publications that aligned with the specified objectives and methodologies”. Did the authors attempt to search for items in Google Scholar or Proquets?

According to your recommendations, we conducted searches in Google Scholar and ProQuest. Based on this search, we found additional source: Quintal C, Lopes J. "Equity in health care financing in Portugal: findings from the Household Budget Survey 2010/2011." Health Economics, Policy and Law. 2016;11(3):233-252

577 - Is a list of abbreviations at the end of the article required?

We have removed the list of abbreviations.

Reviewer 2 Report

Comments and Suggestions for Authors

See enclosed comments.

Comments on the Quality of English Language

See enclosed comments.

Author Response

Response to Reviewers:

Journal: Healthcare

Manuscript No: healthcare-2991471

Manuscript title: Inequalities in out-of-pocket health expenditure measured using financing incidence analysis (FIA): A systematic review.

Corresponding author: Askhat Shaltynov

Referee 1

Changes by the authors

This paper is a systematic review of out-of-pocket health expenditure inequalities in different countries, with exploration of the trends in financing accounting for these differences. Countries more social financing oriented, and those leaning more on private financing are examined. Mention of the role of insurance premiums early on, and whether the authors include co-pays as out-of-pocket expenses, would be helpful. Solid definitions of “progressive financing” and “regressive financing” are necessary. At several points mentioned below, the paper lapses into general jargon that could benefit from greater specificity and an example or two. The need for a more preventive focus should be raised at the end of the Discussion. Grammatical corrections are moderate.

Thank you very much for taking the time to review this manuscript. Please find the detailed

responses below and the corresponding corrections highlighted in the re-submitted files.

P.2, Para. 1, line 48:

Please mention the significance of insurance premiums (your P. 11, line 320) on ability to pay relative to OOPs. Are co-pays part of the OOPs you are citing?

Apologies for the oversight. The text has been corrected to refer to private insurance in the manuscript:

The use of private financing and private insurance schemes often contradicts WHO's goals and financial protection.

OOPs are not part of insurance schemes, as they have separate calculations based on the Kakwani index. We attempted to clarify the concept of post-payment in rural regions of China by modifying the sentence:

In contrast to other modes of healthcare payment, OOP payments in the rural context operated as a post-paid health financing mechanism when healthcare providers may have a financial incentive to utilize costly drugs or advanced technologies under fee-for-service payment models.

P.2, Para. 2, line 50:

Either identify the four challenges for financing, or replace “four challenges” by “several”

Thank you for the feedback. The four challenges identified by Barber et al. have been incorporated into the text in numerical order: 1. Global normative expenditure targets were primarily devised for advocacy purposes, emphasizing the significance of healthcare for national development and securing political commitment; 2. Focusing attention on global normative targets may lead to the mistaken assumption that achieving Universal Health Coverage (UHC) is a fixed threshold or singular, unchanging goal over time; 3. The concept of a global normative target assumes that all countries need to spend a certain amount on healthcare to achieve similar outcomes; 4. Global normative targets typically focus attention solely on funding deficits, leading some policymakers and donors to assume that private financing could fill the gap left by limited government budgetary capacity.

FIA -> financing incidence analysis (FIA)

[True, full spelling appears in title, but must be reiterated in text]

Thank you for the comment. We have replaced the abbreviation in the text with its full form upon its first mention.

P.4, Para. 2, line 156:

Define “regressive financing” and “progressive financing” for the reader unfamiliar with these terms and provide an example of each.

We added a sentence explaining the differences between regressive and progressive systems:

A positive value indicates that the health financing mechanism j is progressive, indicating that wealthier households contribute a higher proportion relative to their share of total household income. Conversely, a negative value suggests that the health financing mechanism is regressive, meaning that poorer households contribute a larger proportion of their income compared to their share of total household income.

P.6, Table 1, Molla et al. citation:

OOP KI, -> OOP KI*,

[To refer to table Legend]

We have added references for the table legend to provide clarification.

P.8, Para. 2, lines 208-210:

2

This is an early opportunity to explain the language being used so the reader can carry this understanding forward. What is a direct versus an indirect tax; direct versus indirect payments (from line 459)? In plain English, what do “a slight progressivity in direct taxes,” a “regressive pattern in the distribution of social insurance” mean? The reader can then use this understanding in subsequent passages.

Thank you for the recommendation. We have added a sentence defining direct and indirect taxes to the text:

 Direct taxes are those imposed directly on individuals or entities, such as income taxes or property taxes, while indirect taxes are imposed on goods and services, like sales tax or value-added tax. Direct payments refer to payments made directly by individuals or entities for healthcare services, while indirect payments refer to payments made indirectly through taxes or insurance premiums.

There isn't a definitive classification of the Kakwani index values, but authors frequently use terms like "slightly regressive" or "slightly progressive" to characterize the index when it is close to 0. Examples of usage:

 Yu CP, Whynes DK, Sach TH. Equity in health care financing: The case of Malaysia. Int J Equity Health. 2008 Jun 9;7:15. doi: 10.1186/1475-9276-7-15. PMID: 18541025; PMCID: PMC2467419.

Ahmed, Y., Ramadan, R. and Sakr, M.F. (2021), "Equity of health-care financing: a progressivity analysis for Egypt", Journal of Humanities and Applied Social Sciences, Vol. 3 No. 1, pp. 3-24. https://doi.org/10.1108/JHASS-08-2019-0040

Crivelli, L., Salari, P. The inequity of the Swiss health care system financing from a federal state perspective. Int J Equity Health 13, 17 (2014). https://doi.org/10.1186/1475-9276-13-17

We have added definition in the brackets: China, Mauritius, South Korea show a slight progressivity (Kakwani index is close to 0) in direct taxes.

We have paraphrased the sentence:

With the exception of South Korea, the selected studies consistently showed a negative value of Kakwani indices in the distribution of social insurance.

P. 11, Para. 4, lines 323-5:

Delete sentence from “Analysis conducted” to “weakly regressive.”

[Lines 316-7 cover this material.]

This sentence was deleted

P. 12, Para. 1, lines 336-7:

Either substantiate or remove this add-on point.

This sentence was deleted

P. 12, Para. 3, line 351:

“redistributing income in the health sector” > Please be more specific.

The sentence has been specified:

redistributing income in the health sector to support low-income groups with the targeted subsidy plan

P. 12, Para. 4:

lines 354-5: “OOP payments for healthcare are regressive” -> Is this comment not circular as it is worded? If regression is something more than undue OOP expenses to lower income individuals, please explain.

Thank you for your feedback. Indeed, frequent use of the term "regressive" may mislead the reader. We have replaced the word "regressive" in the sentence to avoid repetition. However, it's worth noting that the out-of-pocket payment mechanism can also be progressive, as shown in some studies included in this review:

The next study conducted by Rezaei et al. from Iran has found that OOP payments for healthcare are inequitable

lines 359-10: “This suggests that OOP have become more progressive, but that value remains strongly negative” -> Please reword this passage so it is more easily understood by the lightly acquainted reader.

We have reworded this passage:

This indicates that the healthcare financing mechanism through OOP has become fairer over the observation period. However, the Kakwani index values remain negative in the final period of observation, emphasizing the inequality experienced by poor households.

P. 13, Para. 1, line 385:

(Add references)

Apologies for our oversight. We have moved the source citation to the end of the paragraph. Additionally, we replaced the word "pattern" with clearer definitions. The last sentence of this paragraph has been removed as it is more suitable for discussing overall results and conclusions.

The progressive OOP health expenditures mechanism observed in the early years of reform shifted to a regressive after comprehensive insurance policies were implemented. This picture, coupled with poor economic growth, has led to increased healthcare spending for individuals.

P. 13, Para. 3, line 400:

“shift from direct to indirect taxation” -> Please provide an i.e., in parentheses

We have provided additional information in the brackets:

(i.e., this means a reduction in the share of corporate tax and personal income tax in favor of VAT)

P. 13, Para. 3, lines 401-2:

Delete sentence from “OOP are regressive anywhere” to “in the North.” [Lines 393-4 cover this material.]

This sentence was deleted

P. 14, Para. 5, line 459:

For the O’Donnell mention, please bring its citation placement (currently #35) forward a sentence. Then explain why a difference exists in the Sarker et al. and O’Donnell et al. findings. Is the difference illustrative of any specific point?

Thank you for your comment. We have corrected the citations. Regarding the O'Donnell study, it covered the period from 1999 to 2000 and did not assess OOP payments or private payments. We have added this study to show general situation in healthcare financing system of Bangladesh described in previous studies.

P. 15, Para. 3, line 523:

“issues of fairness and policy” too general. Please be more specific, even adding an additional explanatory sentence.

We have changed this sentence with two another sentences to clarify the passage:

In the context of the economic crisis following the COVID-19 pandemic, coupled with the geographic concentration of privately insured individuals in wealthier northern regions, the relative disadvantage of low-income individuals from southern regions may increase due to rising OOP payments. The study authors emphasize the need to increase the share of progressive healthcare financing mechanisms, especially in the context of the COVID-19 pandemic and its economic consequences.

P. 15, Para. 4, line 539:

Remention the shift from infectious to chronic disease (from line 377) and failure to adopt a preventive focus (from line 438).

We have added two obstacles to achieve UHC:

Based on the analyzed research, achieving UHC remains a challenge due to regional disparities, disparities between urban and rural areas, income inequality, the shift the burden from infectious diseases to chronic illnesses, the imbalance between advocacy, prevention, treatment and the additional burden observed during the COVID-19 pandemic

P. 3, Para. 10, line 138:

Map -> The map

using ESRI -> used ESRI

It has been corrected in the text of the manuscript

P. 3, Para. 11, line 140:

Definition Used -> Definitions Used

It has been corrected in the text of the manuscript

P. 3, Para. 12, line 141:

UHC -> Universal health coverage (UHC)

It has been corrected in the text of the manuscript

P. 4, Para. 1, line 145:

CHE -> Catastrophic health expenditures (CHEs)

of the total consumption. -> of the total consumption of healthcare services.

It has been corrected in the text of the manuscript

P. 4, Para. 6, line 173:

its comprehensiveness -> the comprehensiveness   

It has been corrected in the text of the manuscript

P. 4, Para. 8, line 181:

6 studies conducted at this level, specifically focusing -> 6 studies conducted specifically focusing

It has been corrected in the text of the manuscript

P. 9, Para. 1, line 216:

conducted in India. -> was conducted in India.

It has been corrected in the text of the manuscript

P. 10, Para. 3:

line 243: half of the total household expenditure shares -> half the proportion of total household expenditures

line 244: less than 10% of the shares -> less than 10% of this share

It has been corrected in the text of the manuscript

P. 11, Para. 1, line 278:

paid -> was paid

It has been corrected in the text of the manuscript

P. 11, Para. 2, line 286:

were carried -> that were carried

It has been corrected in the text of the manuscript

P. 11, Para. 4, line 325:

Notably, the trend -> The trend

It has been corrected in the text of the manuscript

P. 12, Para. 1, line 334:

issue of fairness -> issue of differentials

[For the term “fairness” to be used to describe the self-employed, further explanation would be needed.]

It has been corrected in the text of the manuscript

P. 13, Para. 2, line 390:

countries -> country

It has been corrected in the text of the manuscript

P. 14, Para. 3:

The Discussion section should not begin with acronyms. Please use the form “out-of-pocket health expenditures (OOP)” for each of these terms.

It has been corrected in the text of the manuscript

P. 14, Para. 6, line 475:

social and private -> both social and private

It has been corrected in the text of the manuscript

P. 15, Para. 2, line 506:

the integration -> and the integration

It has been corrected in the text of the manuscript

P. 15, Para. 3, line 523:

highlights -> highlight

It has been corrected in the text of the manuscript

P. 16, Para. 1:

line 544: on primary data -> on distinctive sets of primary data

line 546: from difference -> from the difference

It has been corrected in the text of the manuscript

P. 16, Para. 2, line 549:

systems heavily reliant -> systems which are heavily reliant

It has been corrected in the text of the manuscript

P. 16, Para. 3:

line 563: under- took -> undertook

lines 563-4: second reviewer -> second reviewers

third reviewer -> third reviewers

line 567: fles. -> files.

It has been corrected in the text of the manuscript

Round 2

Reviewer 2 Report

Comments and Suggestions for Authors

See enclosed comments.

Comments on the Quality of English Language

See enclosed comments.

Author Response

Response to Reviewers:

Journal: Healthcare

Manuscript No: healthcare-2991471

Manuscript title: Inequalities in out-of-pocket health expenditure measured using financing incidence analysis (FIA): A systematic review.

Corresponding author: Askhat Shaltynov

Referee

Changes by the authors

P.4, Para. 7, line 173:

In their online response to this reviewer for P. 4, Para. 2, line 156, the authors have provided adequate definitions (“A positive value indicates …”, “Conversely, a negative value suggests …”) for “regressive financing” and “progressive financing,” but have failed, perhaps due to oversight, to include them in the text. Line 173 after “… values reflect progressive financing.” but before “Kakwani index = C – G, (4)” would be an appropriate place to complete the insertion.

Thank you for your time and consideration in reviewing our submission. Below are the detailed responses and the corresponding corrections highlighted in the resubmitted file:

A positive value indicates that the health financing mechanism is progressive, indicating that wealthier households contribute a higher proportion relative to their share of total household income. Conversely, a negative value suggests that the health financing mechanism is regressive, meaning that poorer households contribute a larger proportion of their income compared to their share of total household income.

P.9, Para. 1, lines 229, 235-6:

The authors were asked to explain the meaning of “a slight progressivity in direct taxes” and “a negative value in the distribution of social insurance.” While it is understandable they would wish to provide an explanation in terms of the Kakwani index, this explanation employs an equal degree of technical language. Please use nontechnical, everyday language to form the explanation for these two passages.

Thank you for the comment. We have added two clarifying sentences:

It means that individuals with higher incomes contribute slightly more in taxes than those with lower incomes.

It suggests that individuals with lower incomes in these countries spend a higher percentage of their money on social insurance compared to those with higher incomes.

P.4, Para. 3, line 154:

Map -> The map

It has been corrected in the text of the manuscript

P.17, Para. 1, lines 569-70:

to increase the share of progressive healthcare financing mechanisms -> to increase the use of progressive healthcare financing mechanisms

It has been corrected in the text of the manuscript
